# Body Image and Lifestyle Behaviors in High School Adolescents

**DOI:** 10.3390/children10071263

**Published:** 2023-07-22

**Authors:** Manon Bordeleau, Natalie Alméras, Shirin Panahi, Vicky Drapeau

**Affiliations:** 1Département D’éducation Physique, Faculté des Sciences de L’éducation, Université Laval, Québec, QC G1V 0A6, Canada; manon.bordeleau.1@ulaval.ca (M.B.); shirin.panahi.1@ulaval.ca (S.P.); 2Centre de Recherche de L’institut Universitaire de Cardiologie et de Pneumologie de Québec-Université Laval, Québec, QC G1V 0A6, Canada; natalie.almeras@criucpq.ulaval.ca; 3Centre Nutrition, Santé et Société (NUTRISS), Institut sur la Nutrition et les Aliments Fonctionnels (INAF), Université Laval, Québec, QC G1V 0A6, Canada; 4Centre de Recherche Interuniversitaire sur la Formation et Profession Enseignante (CRIFPE), Université de Montréal, Québec, QC H3T 1J4, Canada; 5Département de kinésiologie, Faculté de Médecine, Université Laval, Québec, QC G1V 0A6, Canada

**Keywords:** body size evaluation, body mass index, childhood obesity, overweight, self-perception

## Abstract

This secondary data analysis study aimed to examine the associations between 1) body size perception (BSP) and body size dissatisfaction (BSD) and 2) lifestyle behaviors and BSP and BSD in adolescents. The study pooled cross-sectional data from two studies (*n* = 301) performed in adolescents. Weight and height were measured, while lifestyle behaviors and perceived actual and desired body size variables were self-reported. Linear regression analysis assessed the contribution of sex and zBMI to BSP and BSD scores. Pearson’s correlation explored associations between BSP and BSD. Cohen’s effect sizes compared satisfied and dissatisfied adolescents within the underestimators subgroup. A positive association between BSP and BSD scores was observed among girls living with normal-weight and overweight/obesity only (r = 0.26; *p* ≤ 0.001 and r = 0.38; *p* < 0.05, respectively). Underestimators who were satisfied with their body size showed a moderate effect size for a lower zBMI, a small effect size for lower screen time, and higher sleep duration compared to dissatisfied underestimators. Underestimation was associated with more body size satisfaction in adolescent girls with normal weight and overweight/obesity, suggesting a protective effect of underestimation. These findings support the hypothesis that body size satisfaction and underestimation in adolescents is associated with healthier lifestyle behaviors.

## 1. Introduction

Body image, a person’s positive or negative self-perceptions and attitudes regarding their body, may play an important role in adolescent health. Previous research has shown a bidirectional association between a negative body image and psychological well-being, which may be related to the adoption of non-optimal health behaviors such as unhealthy lifestyle and weight control practices in adolescents of both sexes [1,2,3].

Body size perception (BSP) refers to how a person perceives their actual body size. According to previous research conducted in Québec, Canada, 89% of adolescents living with overweight or obesity (OW/OB) inaccurately identified their body size, compared to 61% of those underweight and 11% of those normal weight [4]. Prior research has shown that adolescents who overestimate their body weight are more likely to adopt unhealthy lifestyle behaviors, which may be a risk factor for obesity. For instance, girls who overestimated their body size tended to have higher screen time [5] while boys demonstrated lower milk, cheese, and yogurt consumption [1]. In addition, adolescents who perceive their body weight status to be heavier are at greater risk of weight gain over time, and this effect appears to be mediated by attempts to lose weight [6]. In contrast, body size underestimation among OW/OB adolescents was associated with lower future weight gain in adulthood relative to those who accurately perceived themselves as OW/OB [7]. Moreover, adolescents living with OW/OB who underperceive their body size are more likely to eat vegetables twice or more per day [8] and sleep an average of at least 8 h per school night [8] compared to accurate perceivers. Thus, the overestimation and accurate perception of OW/OB appears to be associated with more at-risk behaviors whereas underestimation is associated with protective and more healthy behaviors.

Body size dissatisfaction (BSD) refers to the negative subjective evaluation individuals have of their own body, which is the discrepancy between their perceived and desired body size. BSD is common in adolescents but can be modulated based on Body Mass Index z-scores (zBMI) and sex [9]. In girls, the desire to reduce body size has been shown to increase based on their body weight, while boys reported a higher desire to be bigger when classified as underweight [10]. A Québec adolescent survey conducted between 2016 and 2017 found that 57% of girls and 54% of boys reported being dissatisfied with their body size [4]. Evidence also revealed that BSD could be associated with poor sleep quality [11] and higher screen time [12], which may lead to weight gain over time [13]. Thus, dissatisfaction with body size is also associated with an unhealthy lifestyle.

Recent studies have examined the relationship between body image variables, including BSP and BSD, among Canadian children [14] and adolescent girls [15]. Our previous research among 269 children (124 boys and 145 girls) aged 6–13 years found that misperception of body size could potentially serve as a protective factor in younger children living with obesity. This misperception was associated with a reduced drive to desire a thinner body [14]. Our subsequent study in 545 adolescent girls participating in FitSpirit, a physical activity intervention for girls, found that body size overestimation and dissatisfaction were associated with more screen time and lower sleep duration [15]. Overall, results from these two studies underline that the relationship between BSP and BSD is modulated by weight status and that the underestimation of body weight may be protective against unhealthy behaviors. Although this research has contributed to the scientific literature, there are a lack of studies investigating the relationship between these variables and lifestyle behaviors specifically among adolescent boys and girls.

Given the high prevalence of body size misperception and dissatisfaction in adolescents, and that both are associated with unhealthy behaviors [6,11,12], it is essential to improve our understanding of the interaction between BSP and BSD as well as the influence of individual factors such as sex and body weight status. Thus, the objective of this study was to (1) investigate the association between BSP and BSD based on sex and body weight status and (2) explore the association between these body image variables and lifestyle behaviors such as screen time, sleep duration, and vegetable, fruit, and dairy consumption. We hypothesized that individual factors such as sex and body weight status would influence the interactions between BSP and BSD and that body size underestimation and satisfaction would be associated with healthier lifestyle behaviors.

## 2. Materials and Methods

### 2.1. Participants

This study sample is based on secondary data analysis from two high school adolescent studies conducted from 2009 to 2011: the baseline data from *Nutriathlon en équipe* (Study 1.) [16,17] and data from *15 ans et la ville devant soi* (Study 2) [18]. Study 1 comprised a sample of 10 classes from grades secondary I and II (grades 7 and 8) selected from three distinct high schools in Québec City, Canada. These classes were randomly assigned to either an intervention group (*n* = 6) or a control group (*n* = 4). Baseline data from both groups were utilized for the purposes of this study. For the second study, data from adolescents attending five secondary schools in Québec City’s metropolitan area were utilized. In both initial studies, the research professional presented and explained the project to the adolescents in class with the use of the assent form. Then, informed consents were sent home, and parents or legal gardians were invited to return the form indicating whether or not they accepted that their child participated in the study. The main inclusion criteria of the two studies were: (1) studying in a French secondary school and (2) having access to the internet for the first study. Participants were included in the present analyses based on the availability of complete data for the dependent variables. The inclusion of participants in our study was not based on specific criteria but rather on the availability of complete data for the dependent variables. While the initial studies involved a convenience sample approach, where schools agreed to participate, the selection of participants for the present posthoc analyses focused on individuals between the ages of 12 and 19. It is important to note that adolescents with a zBMI lower than −2 were excluded from the present analyses (Figure 1). In order to be included in the analyses, we required information on sex, BMI, BSP, and BSD.

The decision to utilize this existing dataset was driven by several factors. Firstly, both studies provided information on BSP, BSD, and lifestyle behaviors among adolescents. Secondly, pooling the data from these studies allowed for a larger sample size, enhancing the statistical power of our analysis.

The initial ethics approval was obtained from the Université Laval Research Ethics Board in April 2011 for Study 1 (#2011-084) and in May 2010 for Study 2 (#2010-055). An agreement with school boards was also obtained in each study. Written informed parental consent and adolescent assent were obtained from all study participants. In the first study, participants provided informed consent for secondary analyses. The second study included secondary outcomes related to the present study objective. All data were anonymized.

### 2.2. Anthropometric Measures

According to standardized methods, trained research assistants measured adolescents’ weight using a bioimpedance balance, while height measurements were taken using a stadiometer [19]. In addition, measurements were performed privately for each student. The zBMI data were calculated using statistical procedures recommended by the World Health Organization. These procedures involve expressing the BMI value in terms of standard deviations, based on the average BMI of adolescents of the same sex and age [20]. Adolescents’ weight status was classified into distinct categories using scores: underweight (zBMI < −2), normal weight (−2 ≤ zBMI < 1), overweight (1 ≤ zBMI < 2), or obese (zBMI ≥ 2) [20]. Sociodemographic data such as adolescent age and sex were obtained through the questionnaires completed by the participant.

### 2.3. Calculation of Body Size Perception Score and Body Size Dissatisfaction Score

The Figural Rating Scale developed by Collins (1991) [21] was used to assess adolescents’ perceived actual and desired body size to determine BSP and BSD scores. This measure consists of seven figures of adolescents of both sexes, ordered sequentially from underweight to obese, with the figure in the middle representing a normal body size. Participants were asked to select the figure that best represented their current appearance (perceived actual body size) and the figure that reflected their desired appearance (desired body size), considering figures of the same sex as themselves.

Based on the procedure outlined by Maximova et al., a corresponding z-score (−3, −2, −1, 0, 1, 2, 3) was assigned to each figure. [22]. Thus, each figure was associated with a weight status where the thinnest figure represents z-scores of −3 and −2 (underweight) and the heavier figures z-scores of 2 (overweight) and 3 (obesity). The BSP score was determined by calculating the difference between the perceived actual body size and the actual body size (zBMI). Following the method previously published [14], adolescents were assigned to one of the three groups according to their BSP scores: “Underestimators” (BSP score < −0.5); “Accurate estimators” (−0.5 ≤ BSP score ≤ 0.5); and “Overestimators” (BSP score > 0.5). A similar procedure was used to define the BSD score (i.e., the difference between desired body size and perceived body size), which was also divided into three groups: desire to reduce body size (BSD score > 0.5); satisfaction with body size (−0.5 ≤ BSD score ≤ 0.5); and desire to increase body size (BSD score < −0.5).

### 2.4. Lifestyle Behaviors

In study 1, the baseline consumption of vegetables/fruits and dairy products was obtained from a web-based platform [16,17]. Under the guidance of a study coordinator, teachers provided students with an explanation of the concept of servings and gave instructions on how to utilize the web-based platform to record servings of fruits, vegetables, and dairy products. Students were asked to log on to the platform at least twice daily and report their consumption. Mean usual consumption was then calculated over five weekdays [16].

In study 2, all lifestyle variables were collected with online questionnaires [18]. The frequency of vegetable/fruit and dairy product consumption per day, week, or month was obtained by using questions taken from the *Canadian Community Health Survey* [23]. No information on serving sizes was requested. Screen time was measured based on a question adapted from the *Quebec Longitudinal Study of Child Development (QLSCD)* [24]: “On average, how many hours per week or weekend do you spend watching television and using a computer?” Response options for this item included: none, 1–5 h, 6–10 h, 11–15 h, 16–20 h, 21–30 h, and 31 h or more. The answers were recorded using the median value of each option. The screen time variable was calculated by adding these two values reported per day (hours per week) + (hours per weekend) divided by 7. Sleep duration was estimated using a slightly modified version of *School Sleep Habits Survey* [25]. Sleep duration was calculated by counting the hours between usual self-reported night bedtime and usual self-reported wake-up time for school days and weekends. The sleep duration variable was calculated by taking a weighted average of the weekday and weekend sleep duration (reported mean weekday sleep time × 5) + (reported mean weekend day sleep time × 2)/7.

### 2.5. Statistical Analyses

The analyses were performed on a convenience sample of adolescents who participated in either study [16,17,18]. As illustrated in the Flowchart (Figure 1), data from 517 adolescents (199 boys and 318 girls) were pooled together. Subsequently, the overweight and obese groups (OW/OB group) were merged to increase statistical power. Thus, 301 participants were retained for analysis: 105 boys and 196 girls or 219 participants with normal-weight and 82 participants living with OW/OB.

Descriptive statistics were carried out per study, and independent t-tests were used to examine potential significant differences in mean scores for each variable between study 1 and study 2. Multiple linear regression adjusted for age and the study was used to identify predictors of BSP and BSD. The multinomial test was used to examine differences by sex and weight status in the prevalence of each categorical variable for BSP (accuracy, underestimation, and overestimation) and BSD (satisfaction, desire to reduce body size, and desire to increase body size). Pearson’s correlation analyses were conducted to explore the relationships between BSP and BSD scores. Considering that the second objective was exploratory and the small sample size for which we had lifestyle behaviors, effect size analysis was performed using Cohen’s *d* factor. The effect size is interpreted as small (>0.2 and <0.5), moderate (≥0.5 and<0.8), and large (≥0.8) [26]. Statistical significance was set at a *p*-value ≤ 0.05. All analyses were performed with SAS OnDemand for Academics (Cary, NC, USA).

## 3. Results

### 3.1. Participant Characteristics

The mean age of the 301 adolescents was 14.1 ± 1.1 years. For the total sample, 73% (*n* = 219) of adolescents were considered normal weight (67% of boys, 76% of girls), 18% (*n* = 55) overweight (22% of boys, 16% of girls), and 9% (*n* = 27) obese (11% of boys, 8% of girls) according to the WHO Reference charts [21] (Table 1). Participants in Study 1 were younger than those in Study 2 (*p* < 0.001; Table 1).

In the whole group, the mean perceived actual body size was significantly lower (*p* < 0.001) compared to the self-reported zBMI (−0.8 ± 1.0 vs. 0.4 ± 1.1). Moreover, the mean desired body size was also significantly lower than the mean perceived actual body size (*p* < 0.001), which indicates that, on average, adolescents had selected a smaller figure than the one with which they identify (−1.2 ± 1.0 vs. −0.8 ± 1.0).

### 3.2. Body Size Perception and Body Size Dissatisfaction

Using the BSP score, results indicated that more than three-quarters (78%) of all adolescents underestimated their body size (underestimators) and 21% had an accurate perception of it, while only 1% thought they were heavier (overestimators). There was a main effect of body weight status on BSP score (*p* < 0.001) and a main effect of sex and body weight status on BSD score (*p* < 0.001). Body size underestimation was more frequent among adolescents living with OW/OB compared to the normal weight subgroup (95% vs. 71%). The difference was statistically significant between the BSP body weight status subgroups (χ^2^ = 19.73, df = 2, *p* < 0.001). The analyses show that 26% of boys and 38% of girls presented accurate body size estimation. Still, the difference between each BSP group by sex was not significant (χ^2^ = 1.18, df = 2, *p* < 0.55). As shown in Figure 2A, the number of adolescents in each category of BSP (accuracy, underestimation, and overestimation) was different based on body weight status (*p* < 0.001), but not sex (*p* = 0.83). The prevalence of underestimation was higher among boys and girls living with OW/OB compared to the normal weight group (97% vs. 63% and 93% vs. 75%, respectively).

With regards to BSD, 57% of the participants reported being satisfied with their body size, while 33% expressed the desire to reduce their body size and 11% to increase it. Figure 2B indicates a main effect of body weight status and sex (*p* < 0.001 and *p* < 0.001, respectively) but no interaction between those two variables. Accordingly, adolescents living with OW/OB more frequently expressed a desire to reduce their body size compared to the normal weight subgroup (60% vs. 23%, respectively; χ^2^ = 39.69, df = 2, *p* < 0.001). Moreover, the prevalence of adolescents in each category of BSD (desire to reduce body size, satisfaction, and desire to increase body size) was also different based on sex (χ^2^ = 29.53, df = 2, *p* < 0.001). Girls reported more desire to reduce body while boys reported more desire to increase body size.

### 3.3. Relationship between Body Size Perception and Body Size Dissatisfaction Scores

Using the whole sample, Pearson’s correlations did not show an association between BSP and BSD scores (*r* = −0.02; *p* = 0.78). Considering the main effect of body weight status and sex on the BSP and BSD scores, additional correlations were performed by subgroups. As presented in Figure 3A, when divided by sex and body weight status subgroups, significant and positive correlations were found between BSP and BSD scores among normal weight and OW/OB girl’s subgroups (r = 0.26; *p* ≤ 0.001 and r = 0.38; *p* < 0.05, respectively). Furthermore, the relationships between BSP and BSD differed significantly between the two subgroups in girls (slope and intercept: *p* < 0.001). There was no observed correlation among boys (Figure 3B).

### 3.4. Body Size Perception, Body Size Dissatisfaction and Lifestyle Behaviors

Considering that both BSP and BSD have been associated with lifestyle behaviors separately and that underestimation is predominant in every subgroup, these outcomes were analyzed separately according to two new subgroups, either “underestimator satisfied” or “underestimator dissatisfied” (Table 2). Compared to the underestimator dissatisfied subgroup, underestimators who are satisfied with their body size present a significantly lower zBMI (0.9 ± 1.0 vs. 0.4 ± 1.0; *p* < 0.001, respectively), as well as a small effect size for lower screen time (4.3 ± 1.8 vs. 3.5 ± 2.5; d = 0.34, respectively) and higher sleep duration (8.7 ± 1.0 vs. 9.0 ± 0.8; d = 0.32, respectively).

## 4. Discussion

The objective of this study was to investigate the association between BSP and BSD based on sex and body weight status and to explore the associations between body image variables and lifestyle behaviors in adolescents. Results indicate that misperception, particularly body image underestimation and dissatisfaction, were common in adolescents and even more prevalent among those living with overweight and obesity. A significant positive correlation was found between BSP and BSD scores among normal weight and OW/OB girls’ subgroups. This suggests that the underestimation of body size is associated with a greater level of body size satisfaction. Moreover, those who were underestimators and satisfied with their body size tended to report less screen time and longer sleep duration, which are healthier lifestyle behaviors.

In this study, underestimation was highly prevalent in our population of adolescents. Our results show that adolescents living with OW/OB were more likely to perceive themselves to be thinner than they were. These findings are consistent with previous studies which found that higher BMI was associated with greater body size underestimation [27,28]. This misperception could be related to the rising rates of OW/OB in Canada [29]. Some studies suggested that frequent exposure to heavier body sizes may lead to a normalization of such sizes, which can contribute to the under-recognition of overweight and obesity status [30]. Moreover, it has been shown that adolescents with friends and parents presenting a larger morphology are more likely to underestimate their body size [31]. It is also well known that adolescents are influenced by the socio-cultural pressures to be fit, and choosing a smaller figure may be based on a desire to be socially acceptable [32].

Furthermore, dissatisfaction with body size was also prevalent in adolescents. Results indicate that body weight status and sex independently influence BSD scores. Adolescent girls and boys and girls in the OW/OB subgroup were more likely to choose a thinner figure than boys or adolescents in the normal weight subgroup, which supports our hypothesis. These results are also concordant with other studies demonstrating this association [9,10]. The high frequency of dissatisfaction may be explained by adolescents living with OW/OB who are often stigmatized [33] and experience teasing, victimization, and bullying [34]. This higher propensity of girls to show a greater desire to reduce body size could be related to girls being more exposed to a thin ideal, while boys are exposed to a muscular ideal [35]. Previous studies suggest that the aspiration to reach sociocultural beauty ideals, reinforced through media, social networks, peers, and parents, could considerably impact BSD [36,37].

Although misperception and dissatisfaction were highly prevalent in adolescents of both sexes, our results revealed that the association between BSP and BSD was only significantly and positively correlated among girls with normal weight and OW/OB. These findings align with our previous research conducted in children [14] and adolescent girls [15], which showed that the relationship between BSP and BSD scores was influenced by weight status. The visual representation of the association between BSP and BSD (Figure 3) enabled us to better understand the association between BSP and BSD. Accordingly, when illustrating the relationship between BSP and BSD, body size underestimation in girls is associated with higher body size satisfaction (see zone −1 to 1, bottom right), suggesting a protective effect of underestimation. This suggests a protective effect against adopting at-risk behaviors or promoting healthier behaviors in adolescents (i.e., adolescents who underestimate and are satisfied with their body size could be less tempted to adopt unhealthy behaviors to control their body weight). Our previous study, which showed that underestimation and satisfaction were associated with less screen time and more optimal sleeping habits, supports this hypothesis [15].

To further explore the hypothesis that underestimation and satisfaction are protective against unhealthy behaviors, we explored lifestyle behaviors in adolescents who were underestimators and satisfied with their body size compared with those who were underestimators and dissatisfied with their body size (i.e., desired to reduce body size). Underestimators satisfied with their body size had a lower zBMI and appeared to have lower screen time and longer sleep duration than the underestimator and dissatisfied group. Although the association between BSP and BSD should be further investigated, it suggests that underestimation and body size satisfaction seem to favor the adoption healthy behaviors. In the long term, this profile could improve body weight management in these adolescents. However, it is difficult to compare these results with other studies since, to our knowledge, no previous studies have explored the combined relationship of BSP and BSD with lifestyle behaviors.

Considering that adolescence is a crucial period for body image development [38], our results suggest that adolescents should evolve in an environment that promotes a positive body image. Positive body image, which is related to body satisfaction and acceptance through the perception of their physical self and thoughts [39], does not necessarily imply that perceived body size needs to be accurate. Further studies are needed to evaluate if accurate BSP is important for children and adolescents to develop a healthy relationship with their bodies and eating behaviors [8]. Understanding adolescent BSP and its relationship with BSD is essential in designing programs that promote the development of a positive (vs. accurate) body image and healthy lifestyle behaviors in adolescents of all sizes.

### Strengths and Limitations

There are several strengths to highlight in this study. Firstly, there was the inclusion of both boys and girls to explore the contribution of sex and zBMI differences to the relationship between BSP and BSD in adolescents, which has been investigated only in children [14] and in adolescent girls [15]. Secondly, actual weight and height were measured through standardized protocols instead of self-reported measurements. Thirdly, as most of the studies had highlighted the relationship between body image variables and healthy lifestyle habits separately [1], this study’s innovative aspect stems from exploring the relationship/interaction between BSP and BSD variables and lifestyle behaviors.

Some limitations need to be underlined. This was a cross-sectional study with self-reported data that does not permit the establishment of a causal association between our variables. Another limitation arises from the utilization of the Figural Rating Scale developed by Collins (1991) [21]. This tool assesses only two dimensions of BSD (the desire to reduce or increase body size) and does not incorporate the aspect of muscularity, which may be relevant to boys’ body size consideration. These findings may be specific to the characteristics and context of the selected location, limiting the generalizability of the results to other populations. Further research with more diverse samples from multiple locations would be beneficial to validate and extend the findings of this study. Additionally, collecting more recent data would enable the comparison of the results with current dynamics and trends in body image perceptions and behaviors among adolescents.

This study has practical implications. Understanding the relationship between body size perception and satisfaction and their association with lifestyle behaviors is important for the development of positive health promotion approaches in adolescents. Our results highlight that promoting a positive rather than an accurate body image in adolescents of all sizes may be essential to favor healthy lifestyle behaviors.

## 5. Conclusions

These results highlight that most adolescents underestimate their body size, which is more prevalent in adolescents who are overweight or living with obesity. In girls, underestimating their body size was associated with more body size satisfaction. This profile could be protective against unhealthy behaviors as suggested by the lower screen time and longer sleep duration observed in those who underestimated their body size, but who were satisfied.

## Figures and Tables

**Figure 1 children-10-01263-f001:**
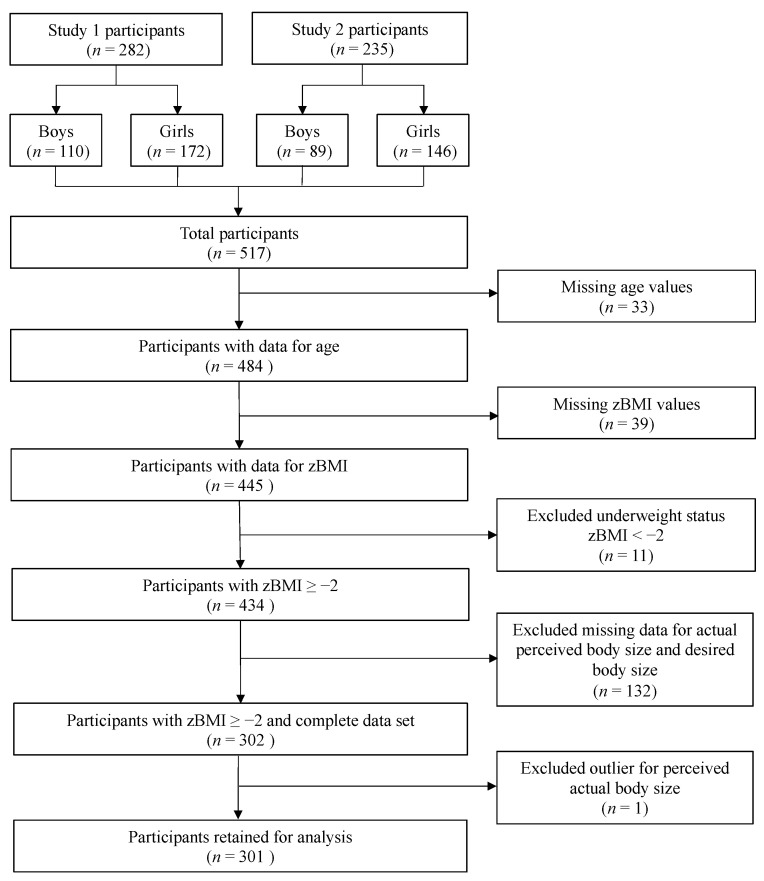
Flowchart of the study selection process.

**Figure 2 children-10-01263-f002:**
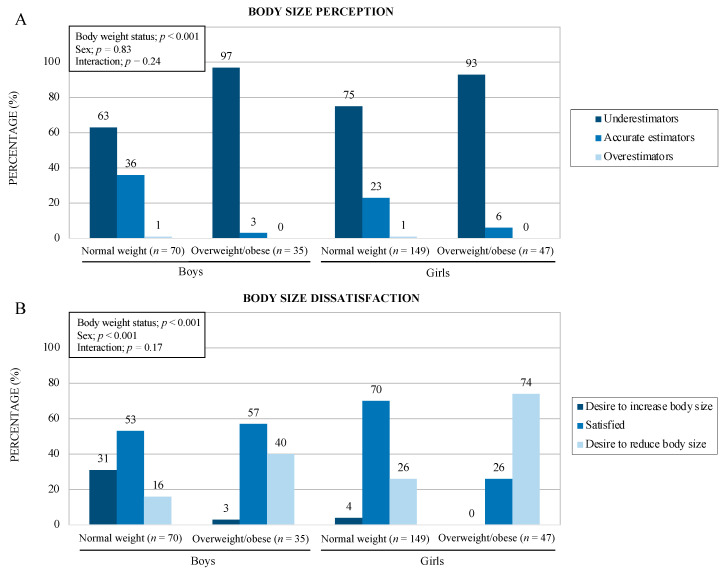
Prevalence of categorization of body size perception (**A**) and body size dissatisfaction (**B**). Legend: Prevalence of categorization of body size perception (accuracy, underestimation, and overestimation) and body size dissatisfaction (satisfaction, desire to reduce body size, and desire to increase body size) by sex (boys and girls) and weight status (normal weight and overweight/obese) (*n* = 301).

**Figure 3 children-10-01263-f003:**
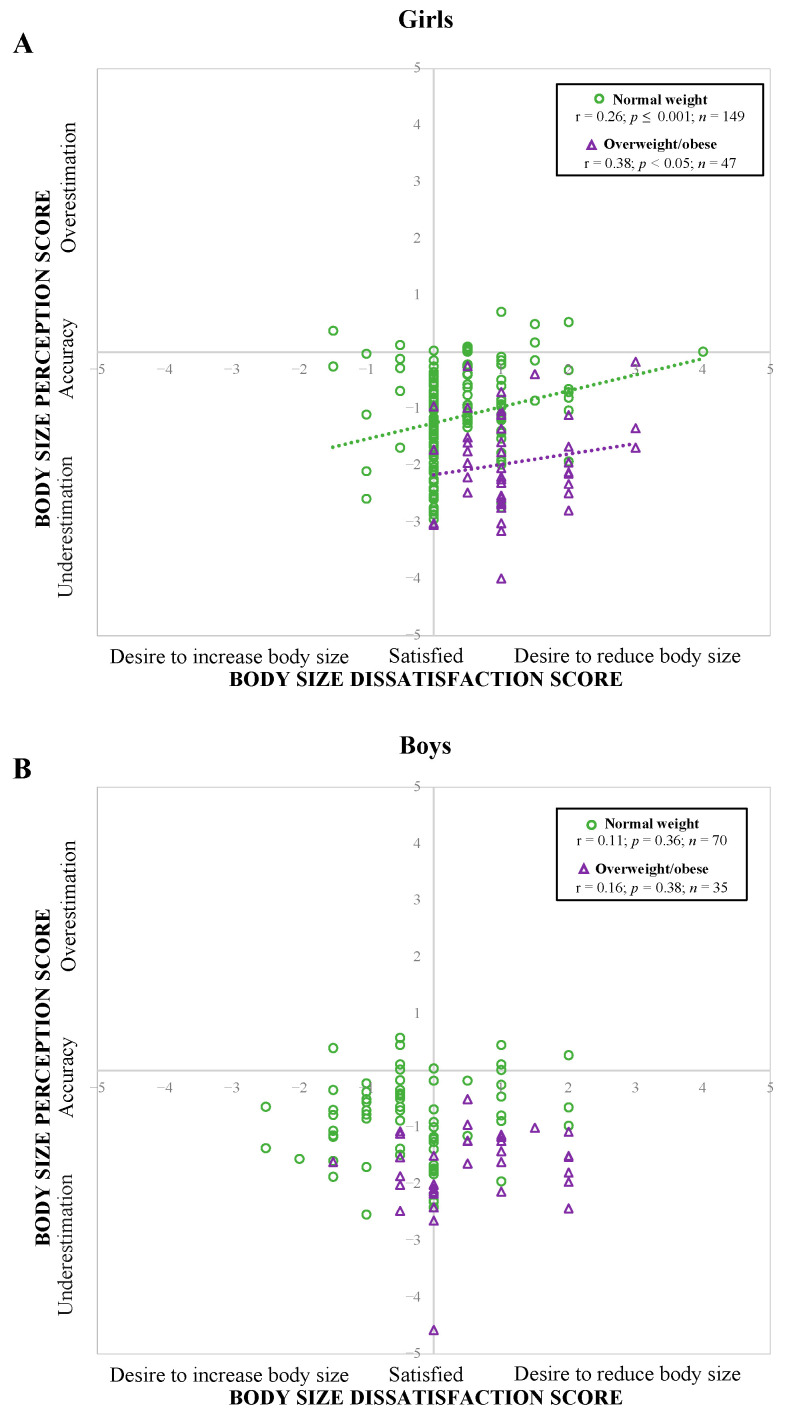
Pearson’s correlation between body size perception score and body size dissatisfaction scores by weight status in adolescents girls (**A**) and boys (**B**). Legend: Pearson’s correlations were adjusted for age and studies.

**Table 1 children-10-01263-t001:** Characteristics of participants classified by study.

	ALL (*n* = 301)	STUDY 1 (*n* = 213)	STUDY 2 (*n* = 88)	*t*-test *p*-Value
**Boys/Girls (%)**	35/65	38/62	28/72	---
**Age (y)**	14.1 ± 1.1 (12.0, 17.0)	13.6 ± 0.8 (12.0, 16.0)	15.3 ± 0.5 (15.0, 17.0)	<0.001
**Body image variables**				
zBMI	0.4 ± 1.1 (−1.98, 3.20)	0.5 ± 1.1 (−1.98, 3.20)	0.4 ± 0.9 (−1.24, 3.08)	0.53
Normal weight Overweight/obese (OW/OB)	73% 27%	69% 31%	82% 18%	---
Perceived actual body size	−0.8 ± 1.0 (−3.0, 2.0)	−1.0 ± 1.0 (−3.0, 2.0)	−0.4 ± 0.7 (−2.0, 1.5)	<0.001
Desired body size	−1.2 ± 1.0 (−3.0, 1.5)	−1.4 ± 1.0 (−3.0, 0.5)	−0.6 ± 0.7 (−2.0, 1.5)	<0.001
Body size perception (BSP) score	−1.3 ± 0.9 (−4.6, 0.7)	−1.5 ± 0.9 (−4.0, 0.6)	−0.7 ± 0.7 (−4.6, 0.7)	<0.001
Body size dissatisfaction (BSD) score	0.3 ± 0.9 (−2.5, 4.0)	0.4 ± 1.0 (−2.5, 4.0)	0.2 ± 0.7 (−1.5, 2.0)	0.09
**Lifestyle behaviors**				
Screen time (h/day)		---	3.5± 2.1 (0.7, 12.5)	---
Sleep duration (h/day)		---	9.0± 0.7 (7.4, 10.3)	---
Vegetables/fruits (serving/day)	(*n* = 224) 3.6 ± 2.3 (0.0, 14.0)	(*n* = 149) 3.3 ± 2.3 (0.0, 11.9)	(*n* = 75) 4.1 ± 2.3 (0.3, 14.0)	<0.05
Dairy products (serving/day)	(*n* = 237) 2.2 ± 1.6 (0.0, 11.0)	(*n* = 149) 1.7 ± 1.2 (0.0, 6.8)	(*n* = 88) 2.9 ± 1.8 (0.0, 11.0)	<0.001

Data are presented as means ± SD (min, max).

**Table 2 children-10-01263-t002:** Characteristics and lifestyle behaviors of participants between underestimators satisfied and underestimators dissatisfied.

	Underestimators Satisfied (*n* = 134)	Underestimators Dissatisfied (*n* = 100)	Cohen’s *d*
Age (y)	14.1 ± 1.1 (12.0, 17.0)	13.8 ± 1.0 (12.0, 16.0)	0.30
zBMI	0.4 ± 1.0 (−1.69, 3.20)	0.9 ± 1.0 (−1.44, 3.01)	0.56
Lifestyle behaviors			
Screen time (h/day)	(*n* = 38) 3.5 ± 2.5 (0.7, 12.5)	(*n* = 12) 4.3 ± 1.8 (1.8, 7.5)	0.34
Sleep duration (h/day)	(*n* = 38) 9.0 ± 0.8 (7.4, 10.3)	(*n* = 12) 8.7 ± 1.0 (7.4, 10.1)	0.32
Vegetables/fruits (serving/day)	(*n* = 98) 3.6 ± 2.4 (0.0, 11.9)	(*n* = 76) 3.3 ± 2.1 (0.1, 9.1)	0.15
Dairy product (serving/day)	(*n* = 107) 2.1 ± 1.4 (0.0, 7.0)	(*n* = 76) 1.9 ± 1.4 (0.0, 6.8)	0.16

Data are presented as means ± SD (min, max); Cohen’s *d* test was performed between subgroups; Cohen’s *d* effect was interpreted as small (>0.2 and <0.5), moderate (≥0.5 and<0.8) and large (≥0.8).

## Data Availability

Data will be available upon request.

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
