# Peer review of "Body Image and Lifestyle Behaviors in High School Adolescents"

_children, 2023, doi:10.3390/children10071263_

Round 1
Reviewer 1 Report
I reviewed this paper in depth, and the results are as follows.
1. In the abstract, the design of this study (secondary data analysis study) and the source of the data should be described.
2. The data used as a secondary analysis in this study were conducted from 2009-2011, and as of June 2023, the time gap is too long. The justification for using this data must be described. It is believed that secondary data analysis with recent data or research comparing the two previous studies applied in this study with the recently measured data is very meaningful in this field.
3. The implications of the two previous studies in this study must be described.
4. The data acquisition process used as an analysis in this study must be described.
5. Since the IRB (Study 1 (#2011-084) & 2 (#2010-055) described in Materials and Methods were not received in this study, IRB should be described for the secondary data analysis study in this study.
6. In the case of the original study, the selection criteria and exclusion criteria of the study participants must be presented.
7. It should be described whether the values and interpretations of zBMI are the same between the original study point and the secondary data analysis point.
8. The Anthropometric measures tool must be described.
9. In terms of limitations, the generalizability of this study should be described.
Author Response
We thank the reviewers for their positive comments on the manuscript. We understand that some points require further clarification and explanation. We hope that this detailed point-by-point rebuttal will satisfactorily address all raised issues.
Answers to Reviewer 1
1. In the abstract, the design of this study (secondary data analysis study) and the source of the data should be described.
Reply: Thank you for your feedback. We have made the necessary revisions to the abstract to include this suggestion. To provide important contextual information for the readers, the abstract has been updated to include details about the study design (secondary data analysis) and the specific studies from which the data was pooled (Nutriathlon en équipe and 15 ans et la ville devant soi).
2. The data used as a secondary analysis in this study were conducted from 2009-2011, and as of June 2023, the time gap is too long. The justification for using this data must be described. It is believed that secondary data analysis with recent data or research comparing the two previous studies applied in this study with the recently measured data is very meaningful in this field.
Reply: Thank you for raising this point. This justification is now found in the methods and limitations sections of the revised manuscript as follows:
Pages 03, line 114:
“The decision to utilize this existing dataset was driven by several factors. Firstly, both studies provided comprehensive information on body size perception, body size dissatisfaction, and lifestyle behaviors among adolescents. Secondly, pooling the data from these studies allowed for a larger sample size, enhancing the statistical power of our analysis.”
Pages 11, line 365:
“Additionally, collecting more recent data would enable the comparison of the results with current dynamics and trends of body image perceptions and behaviors among adolescents.”
3. The implications of the two previous studies in this study must be described.
Reply: Regarding this comment, we are unclear about the reviewer's request. The two previous studies were conducted between 2009 and 2011 by our research team, focusing on similar adolescent populations living in the Québec City’s metropolitan area. These two studies had similar objectives, which was to understand and promote healthy lifestyle habits in adolescents. They assessed similar variables related to body image and lifestyle behaviors. Moreover, they had similar inclusion/exclusion criteria, which allows us to conduct the present secondary analyses.
4. The data acquisition process used as an analysis in this study must be described.
Reply: The involvement of the two researchers in initial research granted them access to the data, (which were anonymized for the research Team), facilitating the opportunity for conducting thorough analyses based on the available data.
We take this opportunity to specify that initial approvals were obtained in 2011 for Study 1 (#2011-084) and in 2010 for Study 2 (#2010-055). In the first study, participants provided informed consent for secondary analyses. The second study included secondary outcomes related to the present study objective. It is worth noting that, in accordance with ÉP3C - Article 2.4, the Research Ethics Board approval is not required for research projects that solely involve the secondary use of anonymized data. For further information, please refer to this link:
Additional information is now found in the revised version of the manuscript as follows:
Pages 03, line 118:
“The initial ethics approval was obtained from the Université Laval Research Ethics Board in April 2011 for Study 1 (#2011-084) and in May 2010 for Study 2 (#2010-055). An agreement with school boards was also obtained in each study. Written informed parental consent and adolescent assent were obtained from all study participants. In the first study, participants provided informed consent for secondary analyses. The second study included secondary outcomes related to the present study objective. All data were anonymized.”
5. Since the IRB (Study 1 (#2011-084) & 2 (#2010-055) described in Materials and Methods were not received in this study, IRB should be described for the secondary data analysis study in this study.
Reply: As described above, the secondary data analysis study utilized pre-existing anonymized data collected from the original studies where participants had already provided their informed consent for the use of their data. Please refer to the description provided in Comment #4.
6. In the case of the original study, the selection criteria and exclusion criteria of the study participants must be presented.
Reply: We thank the reviewer for the valuable feedback and for raising an important point regarding the selection criteria in our study. We have made efforts to provide more clarity regarding the selection criteria in the revised manuscript. The exclusion criteria are also listed in Figure 1 of our manuscript. However, we provided further clarification on the selection and exclusion criteria as followed:
Pages 02, line 94:
“Study 1 comprised a sample of 10 classes from grades secondary I and II (grades 7 and 8) selected from three distinct high schools in Québec City, Canada. These classes were randomly assigned to either an intervention group (n = 6) or a control group (n = 4). Baseline data from both groups were utilized for the purposes of this study. For the second study, data from adolescents attending five secondary schools in Québec City's metropolitan area were utilized.”
Pages 02, line 103:
The main inclusion criteria of the two studies were: 1) studying in a French secondary school and 2) having access to the internet for the first study. Participants were included in the present analyses based on the availability of complete data for the dependent variables.”
7. It should be described whether the values and interpretations of zBMI are the same between the original study point and the secondary data analysis point.
Reply: The zBMI values were calculated using the same established chart that has been in use since 2007. If we were to calculate zBMI today, we would follow the same methodology and utilize the same chart1. Therefore, the values and interpretations of zBMI remain consistent between the original study and our secondary data analysis.
Furthermore, based on the data presented in Figures 1 and 2, it can be also observed that the distribution of zBMI is similar between the complete group before and after the data selection process i.e., participants with complete data outcomes. This similarity is also observed when analyzing the distribution within different cohorts (Figure 3 and 4).
Figure 1. Baseline distribution of zBMI (n=434).
Figure 2. Distribution after the data selection process (n=301).
Figure 3. Baseline distribution of zBMI for a) cohort 1 (n=230) and b) cohort 2 (n=204).
a)
b)
Figure 4. Distribution of zBMI for a) cohort 1 (n=213) and b) cohort 2 (n=88) after the data selection process.
a)
b)
1Onis, M.d.; Onyango, A.W.; Borghi, E.; Siyam, A.; Nishida, C.; Siekmann, J. Development of a WHO growth reference for school-aged children and adolescents. Bull World Health Organ 2007, 85, 660-667.
8. The Anthropometric measures tool must be described.
Reply: Thank you for raising this point. This information is now found in the revised version of the manuscript:
Pages 03, line 126:
“According to standardized methods, trained research assistants measured adolescents’ weight using a bioimpedance balance, while height measurements were taken using a stadiometer [19].”
9. In terms of limitations, the generalizability of this study should be described.
Reply: Thank you for raising this point. This information is now found in the revised version of the manuscript as follows:
Pages 11, line 363:
“These findings may be specific to the characteristics and context of the selected location, limiting the generalizability of the results to other populations.”

Reviewer 2 Report
- Please add some practical implications for the findings of this article.
Author Response
Answers to Reviewer 2
- Please add some practical implications for the findings of this article.
Reply: Thank you for raising this point. This information is now found in the revised version of the manuscript. The practical implication has been added to the discussion section as follows:
Pages 11, line 364:
“This study has practical implications. Understanding the relationship between body size perception and satisfaction and their association with lifestyle behaviors is important to develop positive health promotion approaches in adolescents. Our results highlight that promoting a positive rather than an accurate body image in adolescents of all sizes may be essential to favor healthy lifestyle behaviors.”
Reviewer 3 Report
Thank you for allowing me to review this interesting study. Overall, the study has raised a very interesting point of discussion. I believe that this study has provided novel findings in this area, allowing readers to think more deeply about what is happening around the sport of climbing, even more so with the recent boom in this discipline.
First of all, I would like to share the need to carry out works like the one you present. They are necessary for the advancement of science in the field they study. The objective of the manuscript is clear and consistent. The study has been an interesting reading, it is necessary to know the reality of the sector on which the work emphasizes.
The abstract includes the necessary elements: background with purpose (objective) of the study, methods, results, main conclusions without exaggerating them.
In the introduction, sufficient ordered references of the publications considered key, with significant and sufficient evidence, are indicated. I find the review of the literature really interesting.
Likewise, reasons are highlighted that justify the importance in a broad context and the current state of the subject investigated. The study is clearly defined and indicates the intention and meaning of the work. The objective to be tested in the study is recorded. The text is understandable and makes clear the main objective of the work and the main conclusions.
In relation to the material and methods, say that the study is described in detail. In addition to the methods, the intervention requirements are indicated in sufficient detail.
In general, it is a very interesting manuscript, despite some questions that are suggested to improve the manuscript and the study findings.
There are a couple of minor points if the authors can amend or clarify:
I would like to show that the instrument used is useful and the tables and graphs that are attached to the manuscript help to know the existing reality on the matter.
How was awareness-raising carried out for the participation of the participants? Because I understand that you have had a conversation with the legal guardians of the study participants. How was the selection of the participants carried out? Participants, through what means? Did the same researcher do it? or were several members who carried out this work.
Don't you think that the participation of the sample, being from the same place, may have influenced the results? I would like this appreciation to be included in limitations of the work.
Author Response
Answers to Reviewer 3
- How was awareness-raising carried out for the participation of the participants? Because I understand that you have had a conversation with the legal guardians of the study participants.
Reply: We thank the reviewer for this comment. In both initial studies, the research professional presented and explained the project to the adolescents in class with the use of the assent form. Then, inform consents were sent home and parents or legal gardians were invited to return the form indicating if they accept or not that their child participates in the study. The text has been adjusted to clarify this point on page 03, line 100.
- How was the selection of the participants carried out? Participants, through what means?
Reply: The participants were not selected through specific criteria but rather based on the availability of complete data for the dependent variables. The convenience sample was used to ensure that the study included as many participants as possible. Following the reviewer’s comment, the text has been adjusted as follows:
Pages 03, line 106:
“The inclusion of participants in our study was not based on specific criteria but rather on the availability of complete data for the dependent variables. While the initial studies involved a convenience sample approach, where schools agreed to participate, the selection of participants for the present posthoc analyses focused on individuals between the ages of 12 and 19. It is important to note that adolescents with a zBMI lower than -2 were excluded from the present analyses (Figure 1). In order to be included in the analyses, we required information on sex, BMI, BSP, and BSD.”
- Did the same researcher do it? or were several members who carried out this work.
Reply : We appreciate your feedback regarding the involvement of the principal researchers in each study. In the revised manuscript, we have provided clarification on this matter. V. Drapeau was actively involved in both studies. Also, V. Drapeau and N. Alméras have the same research area and they have a well-established history of collaboration on various projects.
Pages 11, line 380:
N.A., V.D. conceived and designed research, contributed to data collection and utilized baseline data from the Nutriathlon en équipe (Study 1 – V. Drapeau P.I.) and 15 ans et la ville devant soi (Study 2 – V. Drapeau and N. Alméras as co-researchers); M.B., N.A., V.D. analyzed and inter-preted the data. M.B. prepared figures and drafted the manuscript; N.A., V.D., and S.P. edited and revised the manuscript. All authors contributed to reviewing and approving the final version of the manuscript and agree to be accountable for all aspects of the work.
- 4. Don't you think that the participation of the sample, being from the same place, may have influenced the results? I would like this appreciation to be included in limitations of the work.
Reply: Thank you for raising this point. This information is now found in the revised version of the manuscript. We believe that the fact that the participants are from the same place may limit the generalization of the results. This has been added in the following section:
Pages 11, line 362:
These findings may be specific to the characteristics and context of the selected location, limiting the generalizability of the results to other populations. Further research with more diverse samples from multiple locations would be beneficial to validate and extend the findings of this study.”
Answers to Children Assistant Editor Anastasia Yu
Please note that there is a little high repetition rate in your manuscript, please check the attached report and revise accordingly, the highlighted sentences are found to be similar with sentences in previous published papers. The numbers mean which source they were from. You can find the online sources at the end of the report. Could you please kindly revise it during the revision?
Reply: Thank you for bringing this matter to our attention, and we appreciate your guidance in improving the manuscript. We have carefully reviewed the attached report and will make every effort to address the highlighted sections of repetition during the revision process.
We would also like to provide some context for these similarities. One such example is the mention of the "30th European Congress on Obesity (ECO 2023)" in our manuscript. This similarity is expected as we presented a poster based on the findings of our article at this conference. The abstract of this article was therefore published for this conference.
We also have published a similar study in Childhood Obesity titled "Body Size Misperception and Dissatisfaction in Elementary School Children" (Manon Bordeleau, Geneviève Leduc, Claudine Blanchet, Vicky Drapeau, Natalie Alméras, 2021). In this study, we evaluated the same association in children, utilizing similar terminology and methodology. Given the overlapping research focus and methodology between the present article exploring the same association in adolescents and our previous study in children, it is understandable that there may be small similarities in the wording and process.
Besides, we found that there is no approval date of the ethic information in your manuscript. Please kindly add it in your article during the revision and send it to us by replying this email.
Reply: The initial approvals were obtained in April 2011 for Study 1 (#2011-084) and in May 2010 for Study 2 (#2010-055). This has been added to the text (see page 03, line 118).

Round 2
Reviewer 1 Report
This manuscript has been appropriately modified according to the reviewer's comments. Thank you for your hard work.